# Development, piloting and evaluation of an app-supported psychosocial prevention intervention to strengthen participation in working life: a study protocol of a mixed-methods approach

Johannes Stephan ,[1] Jan Gehrmann ,[1,2] Ananda Stullich ,[1] Laura Hoffmann ,[1] Matthias Richter ,[1] for the PE³PP study group

JS and JG contributed equally.

JS and JG are joint first authors.

For numbered affiliations see end of article.

**Correspondence to**
Johannes Stephan;
johannes.stephan2@tum.de

## ABSTRACT

**Introduction** Rates of incapacity to work due to mental disorders have increased in many European countries. The consequences of persistent stress can impact individuals' physical and psychological well-being and gradually develop into chronic stress. Mental disorders or symptoms of burn-out syndrome can have severe consequences. Mental disorders leading to work incapacity significantly burden the health system. Prevention interventions can protect against burn-out, depression, anxiety and other mental health disorders. Digital health is a promising approach to increase the utilisation of effective prevention interventions. This mixed-methods study evaluates a newly developed app-supported psychosocial prevention intervention called 'RV Fit Mental Health' to strengthen participation in working life.

**Methods and analysis** The study uses a three-stage parallel mixed-methods design. This study accompanies the development (stage 1), piloting (stage 2) and evaluation (stage 3) of the new intervention. Within the stages, there is a quantitative as well as a qualitative research strand. Employed persons with an incipient mental disorder will be included. Additionally, experts within the project or connected areas will be included. Quantitative data will be analysed using multifactorial variance analyses in a pre–post design. Qualitative data will be analysed using qualitative content analysis. The study is a comprehensive research approach to investigate the development, piloting and evaluation of an app-supported psychosocial app-based prevention intervention. The rigour of the study will be achieved through data triangulation.

**Ethics and dissemination** All participants will receive detailed study information and give written informed consent before data collection. Ethical approval was obtained from the Technical University of Munich Ethics Committee. All data collection will follow all legislative rules regarding data protection, also following the Declaration of Helsinki. The study results will be disseminated in peer-reviewed journals and presented at international conferences.

**Trial registration numbers** DRKS00030818 and DRKS00033080.

### STRENGTHS AND LIMITATIONS OF THIS STUDY

⇒ A mixed-methods research design is used for the entire study, which allows multiple perspectives to be focused on the same topic, leading to comprehensive results.

⇒ The generalisation of quantitative and qualitative results is challenging but offers various potentials for comparison and contrast regarding other innovative projects.

⇒ The holistic approach integrating qualitative and quantitative data collection offers the potential to analyse the peculiarities of the digital dimension of the intervention.

## INTRODUCTION

Over the past two decades, cases of incapacity to work due to mental disorders have increased in many European countries and represent a significant burden of disease.[1 2] About one in five people has a current mental disorder (18% in Germany, 17.3% in the EU, 18.9% in the USA and 17.6% worldwide).[3–5] In a study commissioned by the German health insurance company 'Techniker Krankenkasse' (TK), 47% of respondents reported that work, school or study caused them to feel stressed. Participants stated that the main stressors were excessive workload, time pressure, rush, interruptions and disruptions.[6] Hogan *et al.* described that long-term workplace stress is often caused by an imbalance between high work demands and the limited personal resources of workers.[7] The consequences of persistent work-related stress can impact individuals' physical and psychological well-being.[8] Excessive demands at work can gradually develop into chronic work stress, whereby symptoms of burn-out syndrome can be consequences.[8] Long-term symptoms of burn-out syndrome increase the risk of mental

disorders such as sleep disorders, depressive symptoms, adjustment disorders and physical health.[8–10] Employee performance, sick leave, absenteeism, accidents and staff turnover are strongly influenced by mental health problems.[9] A relationship can also be shown between burn-out and incapacity to work, as well as early entry into a disability pension.[10]

A study by another German health insurance company, the 'Deutsche Angestellten Krankenkasse', showed that the number of sick leave cases caused by mental health issues has almost tripled over the last 25 years.[11] The health report of the TK showed similar findings: The number of sick days due to mentally caused inability to work more than doubled from 2000 (129 sick days per 100 insurance years) to 2021 (283 sick days per 100 insurance years) in Germany. The main reasons for this increase were depressive episodes and reactions to severe stress and adjustment disorders.[12] Mental disorders cause the most days of incapacity to work at a stretch in Germany.[11 12] Individuals who are permanently unable to work due to health reasons can apply for and receive a disability pension. The German Pension Insurance Association reports a significant increase, by double, in first-time granted disability pensions due to mental disorders in the last 25 years.[13] As a result, in the year 2021, mental disorders (41.7%) became the primary reason for disability pensions, followed by new formations such as cancer (15.3%), musculoskeletal disorders (11.5%) and cardiovascular diseases (9.1%).[13]

According to the Organisation for Economic Co-Operation and Development (OECD), in 2015, the costs of mental disorders in Germany accounted for approximately 4.8% of the gross domestic product.[4] Compared with the OECD-28 average, Germany's expenditures related to mental disorders are 0.7 percentage points above the average; Germany spends proportionally the second most on mental disorders in Europe.[4]

The European Psychiatric Association states that various mental disorders can be prevented through evidence-based interventions by strengthening protective factors or reducing risk factors to promote mental health and disease prevention, for example, through psychoeducation, skills training, stress management or other various therapeutic elements.[14] Consequently, psychosocial prevention interventions can protect against burn-out, depression, anxiety and other mental health disorders.[8 11 15 16] Effective prevention interventions strengthen environmental and individual resources while reducing risk factors.[17]

In Germany, the National Prevention Conference has, therefore, highlighted protecting and strengthening mental health in the workplace as a priority.[18] The German Pension Insurance supports this and acts according to the principle of 'prevention before rehabilitation'.[19] There is a general need to expand attractive evidence-based prevention interventions and improve access to and utilisation of these services.[20]

## Determinants of the use of digital health applications
Digital health is a promising approach to increase the utilisation and quality of prevention interventions by using information and communication technologies to meet health-related needs.[21] Studies have shown that psychosocial aspects can be strengthened during inpatient treatment with the application of various therapies or elements such as individual or group training or psychoeducation.[22–24] In addition, positive outcomes have been demonstrated for digitally supported mental health prevention interventions using a variety of therapeutic or psychoeducational elements.[25–27] Among the digitally supported prevention interventions that were not exclusively related to work, various outcomes were examined in employees, such as work stress,[25] perceived stress,[26] anxiety and quality of life,[27] which are either related to or exacerbated by the work environment. Digital health applications can be used flexibly in terms of time and location and can therefore be integrated into everyday life alongside work.[25–27] Individuals using digital health applications are exposed to various determinants influencing their effective and efficient use.[28 29] One of the crucial determinants seems to be digital health literacy (DHL).[30] DHL encompasses searching for and finding health information from digital sources, comprehending and assessing it, and applying acquired knowledge to address health-specific questions and solve problems.[31] It also encompasses skills and knowledge to interact productively with technology-enabled health tools.[32 33] Other determinants that influence the utilisation of digital health applications include cultural and social factors,[33–35] motivation and interest,[32 33 36 37] and the accessibility and availability of technology.[38–40] Subsequently, there is evidence of a relationship between DHL and adherence to digital health interventions.[41 42]

When investigating the effectiveness of an app-supported psychosocial prevention intervention, it is crucial to collect data on the different determinants that affect using a digital health application. By expanding concepts of DHL to include sociocultural factors, motivation, interest, technology accessibility and availability, a holistic assessment of DHL can be operationalised that incorporates the relevant determinants and allows predictions about the use of a health application.

## Implementation and acceptance of digital health applications
The importance of analysing the acceptance and framework conditions and contexts of digital health applications is emphasised regarding the use of digital health applications. Therefore, it is essential to account for organisational and contextual factors that impact the implementation of innovations in healthcare settings.[43–45] This is particularly important as it has been shown that evidence of effectiveness has not readily led to sustained implementation.[46] Especially, in mental healthcare, the implementation of digital health applications poses various and unique challenges, highlighting the importance of evaluating the effectiveness and factors for successful implementation.[45 47] Various studies highlight that it is necessary to analyse the interrelation between digital intervention and the healthcare context in which

it is implemented.[48 49] Thus, many digital mental health-care interventions show better effects when paired with some therapeutic or doctoral support.[50] It is also highlighted that the perspective of health professionals as well as patients needs to be analysed in the development to enable successful implementation.[51–53] Analysing the implementation of a digital intervention needs to apply a comprehensive approach using quantitative and qualitative data. As the development, piloting and evaluation is an iterative process, there is a need to analyse the different stages, anticipating the various perspectives involved, like users as well as healthcare professionals, but also focus on the contextual factors like organisational routines, established processes and regulatory or policy aspects.[48 54]

## Problem summary

In summary, the following problems were identified: first, stress experienced by individuals related to or exacerbated by the work environment can become chronic, negatively affecting their mental and physical well-being. Consequently, this places an increasing burden on the healthcare systems. Second, prevention interventions can protect against the chronification of stress; however, prevention interventions are rarely taken up in Germany. Third, digitally enabled prevention interventions can increase utilisation. However, it cannot be assumed that different people can use digital health applications equally effectively. Fourth, there are measurement tools for assessing DHL, but existing measurement tools do not account for potential determinants that may impact the effective use of digital health applications. Fifth, regarding developing and implementing a new digital intervention, there is a need to analyse the acceptance and contextual factors.

The study aims to investigate the development, piloting and evaluation of a psychosocial app-based prevention intervention using a comprehensive mixed-methods approach. The intervention explicitly focused on mental health problems related to or exacerbated by the work environment, such as affective disorders, phobic and other anxiety disorders, adjustment disorders, somatoform disorders and burn-out. Its main aim is to improve the participation of those affected in their working life. The study protocol was developed using the 'Standard Protocol Items: Recommendations for international trials' (SPIRIT) checklist.[55]

## METHODS AND ANALYSIS

### Study background

The project 'Development, piloting and evaluation of an app-supported psychosocial prevention intervention' (German acronym PE³PP) is funded under the federal programme 'Innovative Ways to Participate in Working Life—rehapro' of the Federal Ministry of Labour and Social Affairs Germany. The project runs from October 2021 to September 2026 and is led by the German Pension Insurance Central Germany.

This study protocol outlines the planned scientific accompaniment for developing, piloting and evaluating a psychosocial app-supported prevention intervention called 'RV Fit Mental Health' by the Chair of Social Determinants of Health of the Technical University of Munich (TUM). The intervention aims to promote and strengthen individuals' participation in work life. It is a prospective multicentric study. The two central German rehabilitation clinics, Median Klinik Bad Gottleuba and SRH Medinet Burgenlandklinik, will lead in developing and conducting the intervention. Further PE³PP project partners are the health insurance companies 'Allgemeine Ortskrankenkasse Sachsen-Anhalt' (AOK SAN) in the State of Saxony-Anhalt and 'AOK PLUS' in the states of Saxony and Thuringia.

### Study intervention

#### Innovation of the intervention

The psychosocial prevention intervention 'RV Fit Mental Health' aims to improve existing prevention interventions of the German Pension Insurance. The secondary prevention intervention is conceptually innovative, focusing on the psychological, psychosomatic and psychosocial aspects of work-related participation disorders. It is formally innovative in terms of its temporal and methodological design. The duration of the intervention will be 14 weeks. It commences with a 2-week inpatient initial phase in a rehabilitation clinic and a 12-week app-supported training phase. Participants receive intensive therapeutic support during the inpatient and training phases. The proactive recruitment of participants by German Pension Insurance Central Germany and the two health insurances, AOK SAN and AOK Plus, is also promising.

#### The framework of the intervention

The study is multicentric and will be conducted simultaneously at two clinics. 1020 participants are expected to undergo the intervention from September 2023 to September 2026. Participants complete the 2-week inpatient phase in small groups of 10 participants per clinic. Each participant will be assigned a therapeutic support person for both intervention phases. During the inpatient phase, the participant and therapeutic support person collaboratively establish prevention goals, resulting in a therapy plan for the inpatient and training phases.

#### Modules of the intervention

The prevention intervention covers various topics, including psychological stabilisation, self-care, finding meaning and purpose, coping with multiple stressors and increasing digitalisation, achieving work–life balance, and providing 'first aid' on topics such as burn-out, depression, anxiety, sleep disorders and pain. In addition to psychoeducational elements, the intervention includes training in social and emotional competencies, professional and personal stress management, interaction

and communication training, as well as the provision of mindfulness-based and body-oriented therapeutic elements. These substantive topics will be consolidated into modules, conveyed and practised during the inpatient and training phases.

### Didactical realisation of the intervention

During the inpatient phase, therapists conduct the modules in groups. Therapists conducting the modules include psychosomatic physicians, nutritionists, sports and physiotherapists, and psychologists, each possessing diverse expertise in interventions using a digital health application. In this process, all participants attend most of the modules. Additionally, there will be specific modules tailored to the individual needs of the participants. The content of the modules is conveyed and trained through presentations, lectures, exercises and self-development phases.

In the training phase, the participants could flexibly arrange their training in terms of time and location. The training phase will be supported by an app, creating a digital connection between the rehabilitation clinic and participants. The app will contain the training plan designed by the therapeutic support person to achieve the prevention goals. The training plan will include various modules for practice during this phase. The therapeutic support person assigned to the training phase is a psychologist trained in the administrative aspects of using the app.

The app provides various methods and formats for delivering module content, such as exercises, instructions, training and seminars in video and audio formats. In addition, interaction and communication features, including group chats and video call functions, enable participants to engage with their therapeutic support person and other participants. Consequently, the app supports and accompanies the transfer of knowledge and behaviours initiated during the inpatient phase into the participants' daily lives.

During the training phase, each participant will have a coaching session with their therapeutic support person every 4 weeks. Participants also confer with their therapeutic support person the level of encouragement they needed to consistently train in the individual modules, as outlined in the intervention plan, between coaching sessions. This method aims to increase and maintain commitment to participation in therapy. Through the coaching sessions, the therapeutic support person and the participant will (re)evaluate the progress towards achieving the prevention goals. Moreover, group sessions will take place during the training phase, where participants from the same group in the inpatient phase and a therapeutic support person will work on relevant intervention content and exchange experiences within the group. In the 14th week of the prevention intervention, each participant will have an individual concluding session with their therapeutic support person, ending the intervention.

### Design, aims and research questions

The study follows a parallel mixed-methods design integrating quantitative and qualitative data.[56–58] This approach offers the opportunity to highlight multiple perspectives on the same topic and consequently betters comprehension. Mixed-methods research has several methodological challenges, especially regarding study rigour, data triangulation and integration of the various data.[57–59] Data triangulation is a methodological approach that will contribute to the validity of the results by linking qualitative and quantitative data, offering various potentials for analysis through the dynamic and emergent process of data collection and analysis.[60–63] This is especially beneficial in evaluation and implementation research.[64]

The study consists of three stages in which various research questions are answered (figure 1). In stage 1, we will collect qualitative data. Stages 2 and 3 have a qualitative and quantitative research strand with corresponding objectives and research questions.

### Stage 1: development (July 2022–August 2023)

Stage 1 (S1) aims to scientifically accompany the development of the 'RV Fit Mental Health' intervention. The central research question is: what needs to be done to provide an attractive, innovative psychosocial prevention intervention?

In this stage, the different dimensions of the development process and the requirements for the organisational implementation are of interest. This stage consists of a qualitative research strand. Thus, the results have an evaluative character in that they enable the optimisation and adaptation of the needs of the experts involved in planning and conducting the intervention and provide indications for successful cooperation between the project partners.

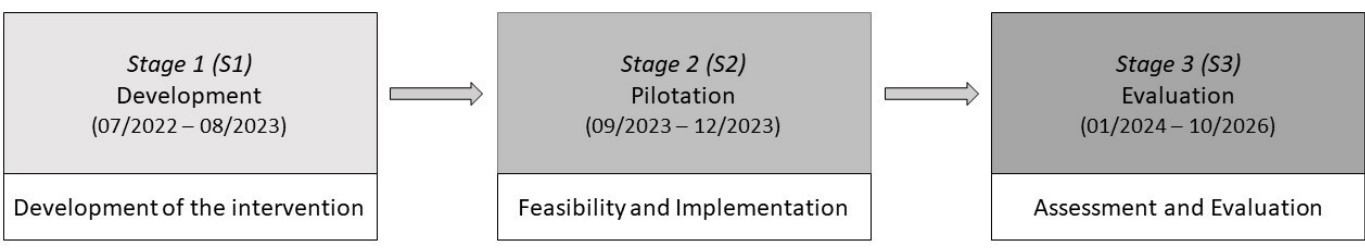

**Figure 1** Overview of the study stages.

### Qualitative research questions

We will answer the following subresearch questions:

► What are the needs and requirements of project partners with regard to the development of psychosocial intervention?

► Does the app fulfil the required functionality for the app-supported training phase?

### Stage 2: piloting (September 2023–December 2023)

In stage 2 (S2), we aim to answer the central research question: How should the psychosocial prevention intervention 'RV Fit Mental Health' be implemented?

We will pilot the intervention and assess its feasibility. Thus, three research objectives emerged for stage 2. First, the needs and experiences of the participants will be analysed to identify standards for addressing potential participants and to optimise the intervention. Second, the constructed instrument for evaluating the intervention will undergo a pretest and a feasibility assessment. As the instrument will include an assessment of DHL, we aim to identify additional skills, competencies and prerequisites that need to be analysed to predict effectiveness. This results in five qualitative and one quantitative subresearch questions for stage 2.

### Qualitative research questions

► How should an app-supported prevention intervention be designed and organised to be feasible and accepted by the participants and the experts?

► What needs, expectations and experiences were the first participants able to perceive during the initial run-through of the prevention intervention?

► What are facilitating factors and barriers regarding the implementation of the intervention?

► Which possibilities for optimisation occur to increase the fit and acceptance of the prevention intervention, especially regarding the fit between the inpatient and the training phase?

► What competencies, skills and prerequisites should users of digital health apps possess besides digital health literacy?

### Quantitative research question

► Is the constructed survey instrument to measure primary and secondary outcomes feasible for participants?

### Stage 3: evaluation (January 2023–April 2026)

Stage 3 (S3) aims to analyse the implementation and execution of the intervention. The key research questions in the evaluation stage are: (1) Is the implementation of the psychosocial prevention intervention achieved? and (2) Is the psychosocial prevention intervention successful?

We will examine organisational conditions, accessibility of the target group and collaboration among the project partners to implement the prevention intervention. We will investigate this from the perspectives of professionals and intervention participants. Additionally, we will assess the effectiveness of the intervention. This results in six qualitative and four quantitative subresearch questions for stage 3.

### Qualitative research questions

► Is the implementation of the psychosocial prevention intervention successful? What facilitating factors and barriers occurred?

► Is the app-supported prevention intervention feasible and accepted by the participants and the experts?

► What are the conditions for the implementing prevention intervention in terms of organisational processes and resources? What are the requirements from the healthcare professional's point of view?

► How is the dovetailing between the inpatient and the digital phase? What are the facilitating factors and barriers in the independent use of the app, and what are the determinants for successful therapeutic support in the digital phase?

► How does the cooperation and collaboration of the different partners develop during the project?

► What are the potentials, challenges and needs for adaptation?

### Quantitative research questions

► What effect does the intervention have on the participants' ability to work, as measured by the Work Ability Index (WAI)?

► What effect does the intervention have on quality of life as measured by the WHO Quality of Life-BREF (WHOQOL-BREF)?

► What effect does the intervention have on perceived stress levels as measured by the Depression Anxiety Stress Scale-21 (DASS-21)?

► Is there a correlation between the level of digital health literacy, measured with the G-eHEALS and a holistic DHL instrument, and postintervention levels of workability, quality of life and perceived stress?

## Intervention participants and study sampling

### Participants in the prevention intervention

There are two pathways through which individuals are recruited for the prevention intervention. Pathway 1 is carried out by the German Pension Insurance Central Germany, a regional institution responsible for two million employed people and over one million pensioners. Pathway 2 is carried out by the health insurance companies AOK SAN and AOK PLUS, regional institutions of the Health Insurance AOK in the east of Germany, insuring a combined 4.3 million citizens. The three insurance companies will contact eligible insureds and proactively suggest the RV Fit Mental Health prevention intervention.

### Sample and recruitment for all study stages

### Qualitative sample: participants of the intervention (S2, S3)

Focus groups (FG) with intervention participants will be conducted during the piloting (S2) and evaluation (S3) stages. We will sample through a random sampling method, where data collection time points will be

predetermined. We plan four FG in the S2 with 6–10 participants each. Ten FG, also with 6–10 participants each, are planned in S3. Overall, we plan up to 14 FG with intervention participants. Contact with the individual group participants for the qualitative data collection is initiated via the respective clinic staff. If there is interest in participation, the clinics forward the time and number of participants. Inclusion criteria are the participation in the prevention intervention and sufficient German language skills.

### Qualitative sample: experts (S1, S2, S3)

Expert interviews with representatives of the German Pension Insurance Central Germany, the AOK SAN, and AOK Plus, and with the executing healthcare professionals from the intervention clinics, will be conducted during the development (S1) and evaluation (S3) stages. There will also be FG in the pilot (S2) stage with these experts. Experts involved in planning and conducting the intervention are included in all stages. Inclusion criteria are the involvement in the development and/or implementation of the intervention, either organisationally or therapeutically. Up to 10 participants will participate in each point of the data collection. The TUM project team will proactively recruit the participants of the expert interviews and the FG.

Furthermore, additional expert interviews will be conducted in S2. Included are healthcare professionals (this may include physicians, psychologists and psychotherapists) who have prescribed or recommended app-based (digital) health applications for disease management, health promotion or prevention as part of a treatment pathway, individuals who are involved in health research regarding the topic of digital health and DHL as well as individuals from health policy in the field of digital health. Excluded are individuals who have prescribed, recommended or are experienced in other digital health technologies, such as wearables (smartwatches, trackers, etc), and people who do not work in healthcare. We will interview 10–12 experts. The experts will be recruited via professional associations and proactively through the study team.

### Quantitative sample: participants of the intervention (S2, S3)

We aim to survey all participants from the intervention's piloting (S2) and evaluation (S3) stages. We will use a non-probability sample for recruitment. Participants in the prevention intervention will receive a flyer on recruitment for our study via the German Pension Insurance and the intervention clinics. On the flyer are instructions and online access for our quantitative data collection. In S2, we expect a total of 60 participants. In S3, we intend to include 960 participants for the pre–post design effectiveness assessment. Inclusion criteria are the participation in the prevention intervention and sufficient German language skills. We expect drop-outs, especially during the digital training phase.

### Data collection
#### Qualitative data collection

The overall qualitative data collection is depicted graphically in figure 2.

The FG with participants occur both during the initial inpatient phase on-site at the clinics and online during the digital training phase. The FG in the inpatient phase occurs towards the end of the inpatient stay. The online FG takes place when at least 4 weeks of the digital training phase have been completed.

| Development<br>(07/2022 − 08/2023) | Pilotation<br>(09/2023 − 12/2023) | Evaluation<br>(01/2024 − 10/2026) |
|---|---|---|
| **Expert interviews**<br>8 interviews | **Expert interviews**<br>10-12 interviews<br><br>**Focus groups with experts**<br>1-2 focus groups (each n=4-5)<br><br>**Focus groups with participants**<br>4 focus groups each with different cohorts − two focus groups (each n=6-10) during the inpatient phase & two focus groups during the training phase (each n=6-10) | **Expert interviews**<br>8-10 interviews<br><br>**Focus groups with participants**<br>10 focus groups (each n=6-10) in an interval of six months with different cohorts of the intervention; evenly divided in inpatient phase as well as training phase |

| **Overview qualitative data collection:**<br>Expert interviews (n=26-30)<br>Expert focus groups (n=4-10)<br>Focus group participants (n=84-140) |
|---|

**Figure 2** Overview of qualitative data collection.

**Figure 3** Overview of quantitative data collection.

The interviews with the experts will be conducted online. Depending on the research questions in each stage, a semistructured interview guide will be developed and pretested for each expert group. The interview guides will be developed following the principles of Helfferich.[65] Furthermore, we are following the methodical principles of expert interviews and recognise the context of the new intervention as well as the evaluative character of the expert interviews.[66–68]

The FG with the experts will take place during a project workshop on-site. The FG is based on a purposive moderation guide starting with a stimulus recognising the specified topics, the organisational contexts and the environment of the intervention.[69]

### Quantitative data collection

Quantitative data will be collected from the study participants in S2 and S3 at four different points (t0–t3) through (online) surveys, depicted graphically in figure 3.

► t0 (first survey) transpires when insured persons are confirmed that participation in the prevention intervention is possible.

► t1 (second survey) takes place at the beginning of the prevention intervention within the first 2 days of the initial inpatient phase.

► t2 (third survey) is conducted 2 weeks after t1, at the end of the initial inpatient phase.

► t3 (fourth survey) occurs 12 weeks after the inpatient phase at the end of the 12-week training phase.

The primary outcomes of interest are workability, assessed using the German Work Ability Index[70]; quality of life, using the German WHO Quality of Life-BREF[71 72]; stress levels, evaluated using the Depression Anxiety Stress Scale-21 (DASS-21)[73–75] and eHealth literacy, which will be assessed using the G-eHEALS[76–78] and a self-developed instrument for holistic DHL. The secondary outcomes are anxiety and depressive symptoms (DASS-21) and self-efficacy using the SWOP K9.[79–82] Further information on the validity and reliability of the assessment instruments is provided in 'online supplemental material 1'. In addition, we are interested in health behaviour as a secondary outcome, which we operationalise as alcohol consumption, smoking, physical activity and dietary behaviour. The training phase behavioural data follow the FITT criteria[83 84]: frequency, intensity, time and type. We will use FITT data to examine participants' adherence to the eHealth application.[85] Additionally, sociodemographic data will be collected during survey t1, and behavioural data from the training phase will be gathered during survey t3.

### Data analysis
#### Qualitative analysis

The semistructured interviews and the FG will be conducted by the research team (JG and JS), audiotaped, verbatim transcribed and analysed following content analysis, using MAXQDA 2022.[86 87] Qualitative content analysis is a systematic, rule-bound approach orientated towards rules of text analysis laid down in advance and contributes to the intersubjectivity of the analysis procedure regarding the defined interests.[87] The analytical steps taken by the category system are, first, the summary and the inductive category formation, then second by the explication and the context analysis, finalised by the structuring and the deductive category formation. The category system will then be rechecked by applying it to the material, the research questions and the case itself.[88] Constant exchange within the study team and the working group for Qualitative Research at the Chair of Social Determinants of Health will ensure the rigour and integrity of the analysis. The analysis will be framed by applying content-analytical quality criteria and reported using the Consolidated criteria for Reporting Qualitative research.[89]

#### Quantitative analysis

Quantitative data analysis will be conducted using the IBM SPSS software (Version 29.0.1.0). During S2, the dataset will undergo a preliminary examination for meaningfulness. This involves checking for the data's completeness, outliers and plausibility. A descriptive characterisation of the quantitative S2 sample will be performed based on sociodemographic data to gain initial insights into the sample.

For S3, we plan a pre–post design effectiveness evaluation. The baseline data will be compared with the data from t1 to t3. For this, we will conduct multifactorial variance analyses to investigate whether there are differences in the groups of individual intervention time points regarding the primary and secondary outcomes, thus

demonstrating the potential effect of the prevention intervention. Additionally, we will perform regression analyses to interpret the relationship between the collected variables, allowing for statements about the influence of individual variables on different outcomes. Similarly, we present a differentiated sample characterisation for S3. In addition, we add a waitlist-controlled component. After collecting data from t0 to t4, we divide participants of S3 into an intervention and waitlist group. We conduct t-tests within each group to analyse the effects between t0 and t1 for the waitlist group and between t1 and t3 for the intervention group. Calculating Cohen's d helps us quantify effect sizes from these tests. We then use a Z-test to compare Cohen's d values of both groups to determine if there are significant differences in effect sizes.

## Patient and public involvement
None.

## ETHICS AND DISSEMINATION
Ethical approval was obtained from the Ethics Committee at the TUM School of Medicine and Health (2022-523 S-SR; 2023-316 S-SB). Following the Declaration of Helsinki, we will conduct the study according to ethical research principles, including the primacy of participant welfare, informed consent (example of participants' declaration of consent as 'online supplemental material 2'), scientific validity, review by an ethics committee, appropriate risk–benefit balance, protection of vulnerable groups, data security and confidentiality, transparent reporting, and voluntary participation. In the inpatient phase, therapists inform participants about the potential disadvantages of app use for the training phase and discuss strategies to mitigate these issues. During the coaching sessions in the training phase, the therapeutic support person systematically identifies any possible disadvantages of using the app. If any study participant decides not to participate or not continue the study, they may withdraw consent at any time without any consequences. All data management follows the requirements of the German version of the General Data Protection Regulation and is approved by the TUM internal department for data security. All data collected in the study will be pseudonymised or, if possible, anonymised. The pseudonymisation through a code ensures data protection of all data. All data containing identificatory characteristics (eg, record, pseudonym assignment list) is secured separately from study data. They are stored in locked rooms at the Chair of Social Determinants of Health. We will delete all data once the study has been completed, but at the latest, after the given legal restriction of 10 years. In case of withdrawal, all data will be deleted entirely or, in extraordinary cases, evaluated carefully anonymised. Deleting data from already analysed or published data is not possible. We will publish all analysed data without any possibility of identifying persons. All results of the quantitative analysis will be published anonymously, and the results of the qualitative analysis will be pseudonymised. The study results will be disseminated in international peer-reviewed journals and presented at international conferences.

**Author affiliations**
[1]Chair of Social Determinants of Health, TUM School of Medicine and Health, Department Health and Sport Sciences, Technical University of Munich, Munich, Germany
[2]Institute of General Practice and Health Services Research, TUM School of Medicine and Health, Department Clinical Medicine, Technical University of Munich, Munich, Germany

**Acknowledgements** The authors thank the PE³PP study group for their support.

**Collaborators** PE³PP study group: Olaf Ballaschke, Sylvia Berlin, Florian Finke, Frank Gabel, Anastasia Grossmann, Claudia Gruhner, Kathleen Kaminski, Andreas Seemann (in alphabetical order).

**Contributors** JG and JS wrote the initial manuscript draft, will collect the quantitative and qualitative data, and are responsible for data analyses. LH and MR were involved in the grant application. JS and JG wrote the study protocol for the ethics review committee. LH, AS and MR critically reviewed the article several times and counselled the entire process of this study protocol and research. All authors have read and approved the final version of the manuscript. The study protocol has been developed on behalf of the PE³PP study group.

**Funding** This study is supported by the Federal Ministry of Labour and Social Affairs as part of the rehapro funding program and is led by the German Pension Insurance of Central Germany (662S0033X1).

**Competing interests** None declared.

**Patient and public involvement** Patients and/or the public were not involved in the design, or conduct, or reporting, or dissemination plans of this research.

**Patient consent for publication** Not applicable.

**Provenance and peer review** Not commissioned; externally peer reviewed.

**ORCID iDs**
Johannes Stephan http://orcid.org/0009-0004-6386-1454
Jan Gehrmann http://orcid.org/0000-0002-5203-016X
Ananda Stullich http://orcid.org/0000-0003-2624-8517
Laura Hoffmann http://orcid.org/0000-0003-4965-6214
Matthias Richter http://orcid.org/0000-0003-3898-3332

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
