## [Reviewer comments · BMJ Open]

ARTICLE DETAILS

TITLE (PROVISIONAL)	Development, piloting, and evaluation of an app-supported psychosocial prevention intervention to strengthen participation in working life: A study protocol of a mixed-methods approach
AUTHORS	Stephan, Johannes; Gehrman, Jan; Stulich, Ananda; Hoffmann, Laura; Richter, Matthias; for the, PE ³ PP study group

VERSION 1 – REVIEW

REVIEWER	Valentim , Olga CINTESIS
REVIEW RETURNED	19-Nov-2023

GENERAL COMMENTS	I would like to thank the editors and authors for the opportunity to review the study protocol "Development, piloting, and evaluation of an app-supported psychosocial prevention intervention: A study protocol of a mixed-methods approach" Digital health is a promising approach to increasing the use of interventions to prevent mental disorders related to persistent occupational stress. In this sense, the present study is important for understanding the effectiveness of an app-supported psychosocial prevention intervention. There are few studies that evaluate this type of interventions, which are fundamental for maintaining the mental health and well-being of workers. In general terms, I can say that study protocol shows scientific integrity and the credibility of the design and methodology. Presents a good bibliographic base, with specific citations from the area of interest, and recent references. I will now offer my contributions or suggestions for improving the text: Introduction The authors justify carrying out the study, including a summary of relevant and current studies, as well as identifying problems related to the study. The literature review is current, selecting only articles/books from the last 5 to 10 years. The objectives are well outlined, and the study protocol was developed using the SPIRIT checklist. Methods and analysis Good description of the various study designs. Page 7, Line 20-21 - the authors state: "During the inpatient phase therapists conduct the modules in groups." I miss that the authors clarify which health professionals could be therapists, what qualifications they have and whether they have training on interventions through a digital health application.
--

	Page 9, line 38 - "...needs of the experts and provides indications for successful cooperation between the actors." It suggested clarifying who the "experts" are and what the inclusion and exclusion criteria are. Page 11, Line 28-39 - in relation to "Quantitative research questions", as they are converted into quantitative assessment, they appear to be open-ended questions. Clarify how these responses will be quantified. Page 13, lines 25-34 - present several instruments to be applied to participants, but do not describe them or whether they are validated for the population studied. This information is suggested, together with reliability and validity data. The inclusion and exclusion criteria for the various participants are unclear. Whether from the people who will be part of the program, or from the experts, or from the health professionals. It is suggested that information/clarification be included on:  a) the inclusion and exclusion criteria for the various participants. b) strategies to improve adherence to the intervention protocol c) plans for data entry, coding, security and storage and when and how to destroy it. d) Composition of the data monitoring committee FINAL DECISION The study protocol needs minor changes. I hope that my contributions serve to improve this study protocol.
--	--

REVIEWER	Csuka, Sára University of Szeged, Institute of Psychology
REVIEW RETURNED	01-Dec-2023

GENERAL COMMENTS	Dear Authors, The topic is highly relevant and carries significant practical implications for mental health applications. Furthermore, the design is well-considered. However, certain modifications could enhance the overall quality of the manuscript:  - The title should specify psychosocial prevention. - I suggest you to revise the abstract to give a more prominent justification for why the study is planned for stress at work - Introduction section: While there may be a reciprocal link, it should be clear that the study focuses on mental problems arising only in connection with work (e.g., burnout) or other mental disorders. - Determinants of the use of digital health applications: *It needs clarification whether these applications are specifically related to work. *It is crucial to substantiate the relevance of work-related issues in this context. How does their usage relate specifically to work? *A distinction between types of work would be pertinent. - Problem Summary: Only work-related stress is highlighted here. Is this of particular importance? Perhaps it should have been emphasized earlier. - As the prevention intervention covers various elements (e.g., trainings, psychoeducation), these should also be included in the
---

	introduction. It would be worthwhile to explain how and why they can be complemented by the application. - 3. Ethics and Dissemination: Ethical considerations could be complemented by addressing potential disadvantages of using the app and explaining how efforts are made to mitigate them. - Regarding data analysis, the text alternately mentions qualitative and quantitative methods. I suggest presenting the qualitative methodology first, followed by the quantitative methodology, to enhance the logical flow of the text. It is recommended to maintain this order consistently in all lists throughout the text.
--	--

VERSION 1 – AUTHOR RESPONSE

Reviewer: 1 - Dr. Olga Valentim , CINTESIS

Comments to the Author:

Comment: I would like to thank the editors and authors for the opportunity to review the study protocol "Development, piloting, and evaluation of an app-supported psychosocial prevention intervention: A study protocol of a mixed-methods approach"

Digital health is a promising approach to increasing the use of interventions to prevent mental disorders related to persistent occupational stress. In this sense, the present study is important for understanding the effectiveness of an app-supported psychosocial prevention intervention.

There are few studies that evaluate this type of interventions, which are fundamental for maintaining the mental health and well-being of workers.

In general terms, I can say that study protocol shows scientific integrity and the credibility of the design and methodology. Presents a good bibliographic base, with specific citations from the area of interest, and recent references.

I will now offer my contributions or suggestions for improving the text:

Introduction

The authors justify carrying out the study, including a summary of relevant and current studies, as well as identifying problems related to the study.

The literature review is current, selecting only articles/books from the last 5 to 10 years. The objectives are well outlined, and the study protocol was developed using the SPIRIT checklist.

Reply: Thank you very much for reading the study protocol and the encouragement for the relevance of our research topic.

Methods and analysis

Comment: Good description of the various study designs.

Page 7, Line 20-21 - the authors state: "During the inpatient phase therapists conduct the modules in groups." I miss that the authors clarify which health professionals could be therapists, what qualifications they have and whether they have training on interventions through a digital health application.

Reply: Thank you for pointing this out. We have updated this information in the manuscript in two places in the subchapter 'Didactical realization of the intervention' as follows:

"Therapists conducting the modules include psychosomatic physicians, nutritionists, sports and physiotherapists, and psychologists, each possessing diverse expertise in interventions using a digital health application."

"The therapeutic support person assigned to the training phase is a psychologist trained in the administrative aspects of using the app."

Comment: Page 9, line 38 - "...needs of the experts and provides indications for successful cooperation between the actors." It suggested clarifying who the "experts" are and what the inclusion and exclusion criteria are.

Reply: Thank you for that remark. In the manuscript, we now refer uniformly to 'project partners' and not to 'actors' and specified the paragraph as followed.

"The results thus have an evaluative character in that they enable the optimization and adaptation of the needs of the experts involved in planning and conducting the intervention and provide indications for successful cooperation between the project partners."

Inclusion and Exclusion criteria for experts (as well as the others samples) are now mentioned in the section 'Sample and recruitment for all study stages'.

"Inclusion criteria are the significant involvement in the development and/or implementation of the intervention, both organizationally or therapeutically."

Comment: Page 11, Line 28-39 - in relation to "Quantitative research questions", as they are converted into quantitative assessment, they appear to be open-ended questions. Clarify how these responses will be quantified.

Reply: Thank you for pointing this out, the quantitative research questions were reformulated and the quantitative measurement instruments for the corresponding outcomes were mentioned in the question. The reformulated research questions in subchapter 'Stage 3: Evaluation' are as follows:

"What effect does the intervention have on the participants' ability to work, as measured by the Work Ability Index (WAI)?

What effect does the intervention have on quality of life as measured by the World Health Organization of Life-BREF (WHOQOL-BREF)?

What effect does the intervention have on perceived stress levels as measured by the Depression Anxiety Stress Scale-21 (DASS-21)?

Is there a correlation between the level of digital health literacy, measured with the G-eHEALS and a holistic DHL instrument, and post-intervention levels of work ability, quality of life, and perceived stress?

Comment: Page 13, lines 25-34 - present several instruments to be applied to participants, but do not describe them or whether they are validated for the population studied. This information is suggested, together with reliability and validity data.

Reply: Thank you for highlighting the importance of providing detailed descriptions of the instruments used in this study. We recognize the necessity of describing these instruments and demonstrating their validation for our study population. While this detailed information is not included in the current manuscript of the study protocol, we assure you that in the manuscript, for the publication of our results, we will include comprehensive descriptions of each instrument along with their respective reliability and validity data. This will ensure the clarity and robustness of our findings. The main reason we did not include the description of the instruments directly in the study protocol is that we have already considerably exceeded the word count. We have asked the editor whether we should, as you suggested, describe the information, reliability, and validity of the instruments in the study protocol, despite exceeding the word count.

Comment: The inclusion and exclusion criteria for the various participants are unclear. Whether from the people who will be part of the program, or from the experts, or from the health professionals. It is suggested that information/clarification be included on:
a) the inclusion and exclusion criteria for the various participants.

Reply: Thank you for the recommendation to provide further information/clarification on the following points a)-d).

Inclusion and Exclusion criteria for all samples are now mentioned in the section 'Sample and recruitment for all study stages'.

Regarding qualitative Sample – Experts: Firstly, for the experts within the project partners:

“Inclusion criteria are the involvement in the development and/or implementation of the intervention, either organizationally or therapeutically.”

And secondly for the additional expert interviews:

“Included are healthcare professionals (this may include physicians, psychologists, and psychotherapists) who have prescribed or recommended app-based (digital) health applications for disease management, health promotion, or prevention as part of a treatment pathway, individuals who are involved in health research regarding the topic of digital health and digital health literacy as well as individuals from health policy in the field of digital health. Excluded are individuals who have prescribed, recommended or are experienced in other digital health technologies, such as wearables (smartwatches, trackers, etc.), and people who do not work in healthcare.”

Regarding qualitative Sample – Participants of the Intervention:

“Inclusion criteria are the participation in the prevention intervention and sufficient German language skills.”

Regarding quantitative Sample – Participants of the Intervention:

“Inclusion criteria are the participation in the prevention intervention and sufficient German language skills.”

Comment: It is suggested that information/clarification be included on: b) strategies to improve adherence to the intervention protocol

Reply: We have added the following supplement in the section 'Didactical realization of the intervention' on how therapy adherence should be supported:

"Participants also confer with their therapeutic support person the level of encouragement they needed to consistently train in the individual modules, as outlined in the intervention plan, between coaching sessions. This method aims to increase and maintain commitment to participation in therapy."

Comment: It is suggested that information/clarification be included on: c) plans for data entry, coding, security and storage and when and how to destroy it.

Reply: Additionally, to the 'Ethics and dissemination' section in the study protocol, we now provide an example of the participant consent form as a 'Supplemental Material' file containing all the mentioned aspects.

Comment: It is suggested that information/clarification be included on: d) Composition of the data monitoring committee

Reply: The newly provided participant consent form contains all relevant information regarding the persons involved in data monitoring.

Reviewer: 2 - Dr. Sára Csuka, University of Szeged, Semmelweis University
Comments to the Author:

Comment: Dear Authors,
The topic is highly relevant and carries significant practical implications for mental health applications. Furthermore, the design is well-considered. However, certain modifications could enhance the overall quality of the manuscript:

- The title should specify psychosocial prevention.

Reply: We suggest two alternatives to the title. We have requested the editor to decide which title is more suitable. The first suggestion is the original title, which we prefer:

"Development, piloting, and evaluation of an app-supported psychosocial prevention intervention: A study protocol of a mixed-methods approach"

The second title contains more specific information and reads as follows:

"Development, piloting, and evaluation of an app-supported psychosocial prevention intervention to strengthen participation in working life: A study protocol of a mixed-methods approach".

Comment: I suggest you to revise the abstract to give a more prominent justification for why the study is planned for stress at work

Reply: Thank you for your suggestion. We have made changes to the abstract, which makes the aim of the intervention, strengthening participation in working life, more prominent.

Comment: Introduction section: While there may be a reciprocal link, it should be clear that the study focuses on mental problems arising only in connection with work (e.g., burnout) or other mental disorders.

Reply: Thank you for your comment. We have added the following sentence to the manuscript to explain the context in which the intervention is focused on mental health problems related to or exacerbated by the working environment in the section of 'Problem summary'.

“The intervention explicitly focuses on mental health problems related to or exacerbated by the work environment, such as affective disorders, phobic and other anxiety disorders, adjustment disorders, somatoform disorders, and burnout. Its main aim is to improve the participation of those affected in their working life.”

Comment: Determinants of the use of digital health applications:

*It needs clarification whether these applications are specifically related to work.

*It is crucial to substantiate the relevance of work-related issues in this context. How does their usage relate specifically to work?

+*A distinction between types of work would be pertinent.

Reply: Thank you for your comments. These applications are not primarily related to work but are being investigated, for example, to determine whether they can reduce work stress, perceived stress, anxiety, and quality of life. The publications did not describe the types of work performed by employees. Digital health applications have the advantage that they can be used anywhere and at any time. This gives the flexibility to integrate the applications into everyday life. We have added a sentence to clarify your points in the 'Determinants of the use of digital health applications' section.

“Among the digitally supported prevention interventions that were not exclusively related to work, various outcomes were examined in employees, such as work stress [25], perceived stress [26], anxiety, and quality of life [27], which are either related to or exacerbated by the work environment. Digital health applications can be used flexibly in terms of time and location and can therefore be integrated into everyday life alongside work [25-27].”

Comment: Problem Summary: Only work-related stress is highlighted here. Is this of particular importance? Perhaps it should have been emphasized earlier.

Reply: Thank you for pointing this out. Work-related stress is not of particular importance at this point, so we have adapted the first two problems of the 'Problem summary' in the manuscript as follows.

“First, stress experienced by individuals related to or exacerbated by their work environment can become chronic, negatively affecting their mental and physical well-being. Consequently, this places an increasing burden on the healthcare systems. Second, prevention interventions can protect against the chronification of stress; however, prevention interventions are rarely taken up in Germany.”

Comment: As the prevention intervention covers various elements (e.g., trainings, psychoeducation), these should also be included in the introduction. It would be worthwhile to explain how and why they can be complemented by the application.

Reply: Thank you for that remark. We included remarks on the various elements covered and referred to the digital health application in that manner:

“The European Psychiatric Association states that various mental disorders can be prevented through evidence-based interventions by strengthening protective factors or reducing risk factors to promote mental health and disease prevention, e.g. through psychoeducation, skills training, stress management or other various therapeutic elements [14].”

“Studies have shown that psychosocial aspects can be strengthened during inpatient treatment with the application of various therapies or elements such as individual or group training or psychoeducation [22–24]. Additionally, positive outcomes have been demonstrated for digitally supported mental health prevention interventions using a variety of therapeutic or psychoeducational elements [25–27].”

Comment: 3. Ethics and Dissemination: Ethical considerations could be complemented by addressing potential disadvantages of using the app and explaining how efforts are made to mitigate them.

Reply: Thank you for your comment, which has helped us to add the planned approach to address potential disadvantages of app use in the 'Ethics and dissemination' section as follows:

“In the inpatient phase, therapists inform participants about the potential disadvantages of app use for the training phase and discuss strategies to mitigate these issues. During the coaching sessions in the training phase, the therapeutic support person systematically identifies any possible disadvantages of using the app.”

Comment: Regarding data analysis, the text alternately mentions qualitative and quantitative methods. I suggest presenting the qualitative methodology first, followed by the quantitative methodology, to enhance the logical flow of the text. It is recommended to maintain this order consistently in all lists throughout the text.

Reply: Thank you for your recommendation, which we implemented in our manuscript. For a logical reading flow, the individual sections presented the qualitative method first, followed by the quantitative method.